# Effect of transcranial direct-current stimulation on cognitive function in stroke patients: A systematic review and meta-analysis

Ru-bing Yan[1☯], Xiao-li Zhang[2☯], Yong-hong Li[3], Jing-ming Hou[1], Han Chen[1], Hong-liang Liu[1] *

1 Department of Rehabilitation, Southwest Hospital, PLA Army Medical University, Chongqing, China,
2 Medical Journal of Chinese PLA, Beijing, China, 3 Department of Evidence-Based Medicine and Clinical Epidemiology, West China Hospital, Sichuan University, Chengdu, China

☯ These authors contributed equally to this work.
* liuhongliangkf@163.com

**Data Availability Statement:** All relevant data are within the manuscript and its Supporting Information files.

## Abstract

### Objective

Transcranial direct-current stimulation (tDCS) is a noninvasive approach that can alter brain excitability. Several studies have shown the effectiveness of tDCS in improving language and movement function in stroke patients. However, the effect of tDCS on cognitive function after stroke remains uncertain.

### Methods

We searched Medline, Embase, the Cochrane Central Register of Controlled Trials (CENTRAL), the China National Knowledge Infrastructure, the China Science and Technology Journal Database, and the Wanfang Data Knowledge Service Platform from inception to April 2, 2019. Two reviewers independently screened the studies, extracted the data, and evaluated the quality of the included studies using the Cochrane Collaboration Risk of Bias Tool. All statistical analyses were performed in RevMan 5.3, and the mean difference (MD) or standard mean difference (SMD) were used as the pooled statistics.

### Results

Fifteen studies involving 820 participants were included. When compared with passive tDCS, anodal tDCS was associated with improved general cognitive performance as examined by the Minimum Mental State Examination or Montreal Cognitive Assessment (SMD = 1.31, 95% CI 0.91–1.71, $P < 0.00001$), attention performance (SMD = 0.66, 95% CI 0.11–1.20, $P = 0.02$). There was no significant difference in memory performance (SMD = 0.41, 95% CI -0.67–1.50, $P = 0.46$).

### Conclusions

tDCS is likely to be effective for patients with cognitive impairment after stroke. The evidence for different effects based on population characteristics and stimulation methods was

**Funding:** This study was supported by the National Key Research and Development Project (2017YFC138503), the National Nature Science Foundation of China (grant no.81671211; 81672251), and the Clinical Innovation Foundation of Southwest Hospital (SWH2016ZDCX4203). The funders had no role in study design, data collection and analysis, decision to publish, or preparation of the manuscript.

**Competing interests:** The authors have declared that no competing interests exist.

limited, but a real effect cannot be ruled out. More high-quality research in this field is required to determine the potential benefits of tDCS in the treatment of cognitive deficits after stroke and to establish the optimal treatment program.

## Introduction

Although stroke has fallen from the second leading cause of death to the fourth in the United States, it remains the leading cause of severe adult disability, which produces a major burden to society [1]. In 2010, there were an estimated 11.6 million events of incident ischemic stroke and 5.3 million events of incident hemorrhagic stroke, most of which were in low- and middle-income countries [2]. In addition to the high morbidity and mortality, the burden of stroke-related disability is another major problem in survivors; the incidence of poststroke cognitive impairment (PSCI) ranges from 22% to 47% in different studies [3–5], and it has had a serious impact on both the economy and quality of life.

Transcranial direct-current stimulation (tDCS) was first developed for clinical purposes in 2000[6]; currently, it constitutes a promising method for neurological condition regulation [7, 8]. As a neuromodulatory approach, tDCS works by depolarizing or hyperpolarizing neuronal membrane potentials through the activation of sodium- and calcium-dependent channels and NMDA receptor activity, thereby modulating neural activity and cortical excitability [7, 9].

Several systematic reviews have evaluated the efficacy of tDCS on motor function and aphasia after stroke [10–12]; some preliminary studies have shown beneficial effects of tDCS on cognitive function in healthy subjects as well as in stroke patients [13–16]. However, it remains largely uncertain whether tDCS promotes the recovery of cognitive function after stroke. Therefore, we conducted this systematic review and meta-analysis to evaluate the effectiveness of tDCS on cognition after stroke.

## Materials and methods

### Protocol and registration

This review adheres to the Preferred Reporting Items for Systematic Reviews and Meta-Analyses (PRISMA) statement (S1 Table [17]). The protocol was published in the International Prospective Register of Systematic Reviews (PROSPERO) on 20 September 2019 (CRD 42019137191).

### Literature search strategy

The Medline, EMBASE, Cochrane Central Register of Controlled Trials (CENTRAL), China National Knowledge Infrastructure (CNKI), China Science and Technology Journal Database (VIP), and Wanfang Data Knowledge Service Platform (WANGFANG) databases were searched from the inception date to 2 April 2019. Appropriate free terms combined medical subject headings (MeSH) was used as the retrieval strategy: "stroke", "transcranial direct current stimulation", "tDCS". Only English or Chinese language articles were included (S2 Table). Relevant reviews and reference lists of all articles were examined for potentially eligible studies.

### Selection criteria

To determine eligibility, two authors (Ru-bing Yan and Xiao-li Zhang) independently reviewed the abstract of each article for the initial selection. Articles that met the following inclusion criteria were retained for full-length text examination: 1) randomized controlled

trials (RCTs); 2) the participants were poststroke patients with cognitive deficits; 3) the intervention and the control group were anodal tDCS versus passive stimulus (sham tDCS or no additional intervention); and 4) the primary outcomes were general cognitive mental status after treatment assessed by Minimum Mental State Examination (MMSE) or the Montreal Cognitive Assessment (MoCA), and the secondary outcomes included attention and memory performance. The exclusion criteria were other study designs, other treatments and studies with incomplete outcome data. Full articles were then screened a second time against the selection criteria to reconfirm eligibility.

## Outcome measures

The general cognitive state of the participants was evaluated before and after treatment using the MMSE and MoCA testing instruments. The MMSE included 30 items covering orientation, memory, attention, numeracy, recall and language skills, and each item received 1 point for accuracy [18]. The MoCA is also a 30-point scale with 7 cognitive subtests: visuo-executive, naming, attention, language, abstraction, delayed recall, and orientation [19]. Both scale scores were positively correlated with cognitive ability. In terms of attention and memory performance, other testing methods, such as the Computerized Neuropsychological Test (CNT), the Loewenstein occupational therapy cognitive assessment (LOTCA) and nonstandard approaches, have also been used.

## Data extraction

Two review authors (Ru-bing Yan and Xiao-li Zhang) independently extracted data from the selected full-text studies using a form created a priori. The following data were extracted: study information (first author, publication year, country, sample size); patient characteristics (age, sex, stroke type, disease onset); interventions (treatment, dose, duration); and outcome data.

## Quality assessment

The Cochrane Collaboration Risk of Bias Tool was used to assess trial quality, and we resolved disagreements by reaching a consensus through discussion [20]. The evaluation of quality was based on the following 7 dimensions: random sequence generation, allocation concealment, blinding of the participants and personnel, blinding of outcome assessment, incomplete outcome data, selective outcome reporting, and other bias.

## Statistical analysis

For all outcomes were continuous variables, we entered the means and standard deviations. The mean difference (MD) and 95% confidence interval (CI) was calculated as a pooled estimate. If the outcome did not use the same unit across studies, we used the standardized mean differences (SMD) instead of the MD. Between-study heterogeneity was assessed by the chi-square test (test level is $P = 0.1$) and $I^2$ statistics, with $I^2 > 50\%$ indicating moderate heterogeneity and $I^2 > 75\%$ indicating severe heterogeneity. Pooled results were calculated with a random model when heterogeneity was significant ($I^2 > 50\%$) [21]. All statistical comparisons were performed in Review Manager 5.3 (http://www.ims.cochrane.org/revman/)[22].

## Subgroup analysis

Subgroup analyses were prespecified on the following factors: 1) types of stroke (infarction or hemorrhage); 2) comparators (sham tDCS or no tDCS) and 3) poststroke duration. Sensitivity analyses were performed to explore the robustness of the results using different statistical

models, different effect measures, and excluding studies with high heterogeneity. Publication bias was assessed using Egger's test and funnel plots in STATA 13.0 [23]. If the *P*-value of Egger's test was 0.1 or lower and the funnel plots appeared asymmetric, a publication bias was indicated.

## Results

### Study selection

The results of the search are summarized in Fig 1. We identified a total of 949 unique records from the inception of each database to April 2019. After screening the titles and abstracts, we excluded records and obtained the full texts of the remaining 49 studies. After further full-text assessment, we determined that 15 studies with 16 trials met the inclusion criteria [24–38]; one of the studies contained 2 independent trials [16]. Two studies were excluded from the meta-analysis since there were no available data [24, 38].

### Study characteristics

The main characteristics of the 15 studies are presented in Table 1. Of the 820 participants, 411 were in the anodal tDCS group and 409 in the control group; approximately 512 (62.4%) were male, and 668 (81.4%) had an infarction stroke. The mean age of the patients ranged from 53.1 to 68.5 years. Seven trials included both stroke types, seven studies focused on ischemic stroke alone and one study focused on hemorrhagic stroke alone. It should be noted that, in all studies, tDCS was administered in combination with traditional cognitive rehabilitation programs or medicine rather than tDCS alone. We found that most of the clinical trials were conducted in China and Korea, and there were no studies from North America or Europe.

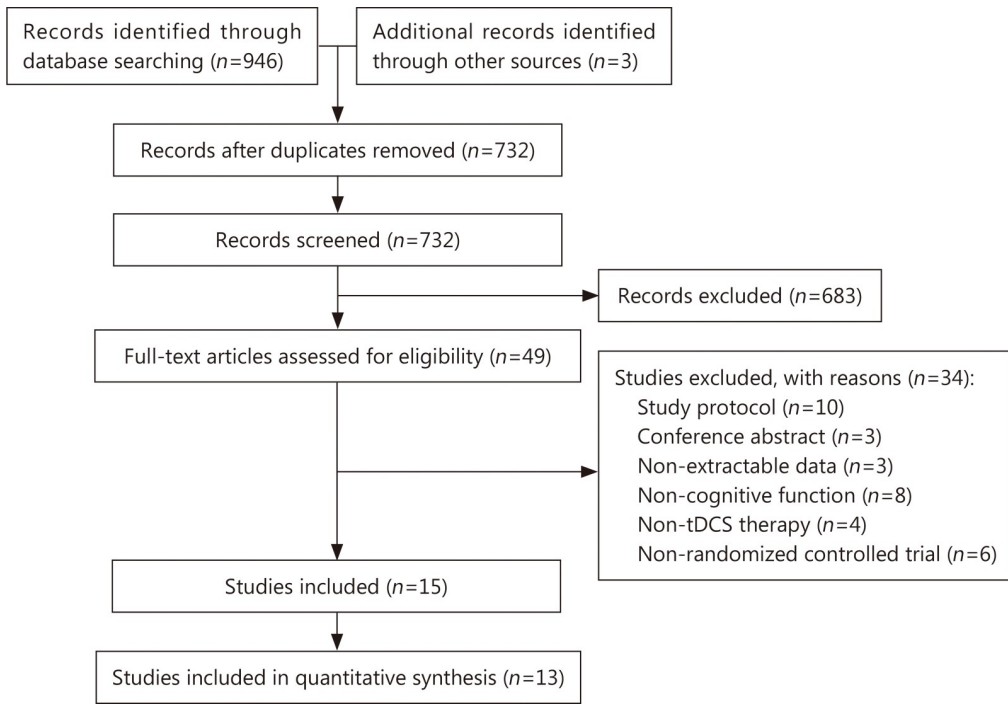

**Fig 1. PRISMA flow diagram.**

**Table 1. Characteristics of the included studies and patients.**

| Study | Design | Intervention | | | Male/ Female | Ischemic/ Hemorrhage | Ages | Onset | MMSE | Follow-up |
|-------|--------|-----------|----------|-----------|------|------|------|------|------|------|
| | | Treatments | Location | Intensity | | | | | | |
| Park 2013 | RCT | anodal tDCS + CRP | prefrontal | 2 mA, 30 min, 5 times/w | 4/2 | 4/2 | 65.3±14.3 | 29.0±18.7 d | 8.2±3.4 | 18.5 d |
| | | sham tDCS + CRP | | | 3/2 | 4/2 | 66.0±10.8 | 25.2±17.5 d | 10.2±2.6 | 17.8 d |
| Yun 2015 | RCT | anodal tDCS + CRP | frontotemporal | 2.0 mA, 30 min, 5 times/w | 6/9 | 7/8 | 60.9±12.9 | 42.2±31.9 d | 20.1±4.8 | 3 w |
| | | sham tDCS + CRP | | | 7/8 | 10/5 | 68.5±14.6 | 39.5±29.6 d | 19.0±5.2 | |
| Shaker 2018 | RCT | anodal tDCS + CRP | dorsolateral prefrontal | 2.0 mA, 30 min, 3 times/w | 20/ | 20/0 | 54.45 ± 4.68 | 14.05 ± 1.53 m | 19–24 | 4 w |
| | | sham tDCS + CRP | | | 20/ | 20/0 | 53.05 ± 6.32 | 16.55 ±2.78 m | 19–24 | |
| Zeng 2019 | RCT | anodal tDCS + CRP | dorsolateral prefrontal | 2.0 mA, 20 min, 5 times/w | 9/6 | 15/0 | 56.21±9.11 | 41.29±10.37 d | 19.71 ±3.65 | 4 w |
| | | sham tDCS + CRP | | | 11/4 | 15/0 | 53.14±7.12 | 43.36±12.17 d | 15.07 ±1.6 | |
| Chen 2019 | RCT | anodal tDCS + CRP | prefrontal | 1.2 mA, 20 min, 5 times/w | 16/24 | 28/12 | 55.19±6.62 | 2.86±1.28 m | <27 | 6 w |
| | | no tDCS + CRP | | | 15/25 | 32/8 | 56.41±6.24 | 2.75±1.39 m | | |
| Guo 2015 | RCT | anodal tDCS + CRP | prefrontal | 1.0 mA, 20 min, 6 times/w | 19/11 | 16/14 | 58.33±9.26 | <4 w | - | 4 w |
| | | no tDCS + CRP | | | 20/10 | 19/11 | 59.59 ±10.02 | | - | |
| Jiang 2019* | RCT | anodal tDCS + CRP | affected frontotemporal | 0.5 mA, 20 min, 5 times/w | 19/17 | 36/0 | 60.3±4.3 | 17.3±5.5 h | - | 12 w |
| | | no tDCS + CRP | | | 20/16 | 36/0 | 61.5±4.2 | 16.7±5.3 h | - | |
| Jiang 2019* | RCT | anodal tDCS + CRP | affected frontotemporal | 0.5 mA, 20 min, 5 times/w | 14/10 | 0/24 | 61.4±4.1 | 15.5±5.7 h | - | 12 w |
| | | no tDCS + CRP | | | 13/11 | 0/24 | 60.7±4.2 | 16.1±5.3 h | - | |
| Luo 2019 | RCT | anodal tDCS + CRP | affected frontotemporal | 1.8 mA, 20 min, 5 times/w | 18/14 | 18/14 | 56.01±6.98 | 76±29.98 d | 13.98 ±3.9 | 6 w |
| | | no tDCS + CRP | | | 17/15 | 17/15 | 55.85±7.02 | 78±29.56 d | 14.01 ±3.9 | |
| Song 2019 | RCT | anodal tDCS + CRP + M | prefrontal | 1.0 mA, 20 min, 6 times/w | 15 | 17/13 | 64.76±8.65 | 45.52±8.62 d | - | - |
| | | no tDCS + CRP + M | | | 15 | | | | | |
| Sun 2016 | RCT | anodal tDCS + CRP | prefrontal | 1.5 mA, 20 min, 6 times/w | 17/9 | 26/0 | 56±9 | 37±16 d | - | 4 w |
| | | no tDCS + CRP | | | 19/8 | 26/0 | 57±8 | 30±16 d | - | |
| Tong 2019 | RCT | anodal tDCS + CRP + M | prefrontal | 1.1 mA, 20 min, 10 times/w | 18/13 | 31/0 | 64.7±7.3 | 40.1±11.5 d | 18.5±2.6 | 4 w |
| | | no tDCS + CRP + M | | | 20/11 | 31/0 | 64.4±7.9 | 36.7±13.4 d | 17.6±2.9 | |
| Tong 2018 | RCT | anodal tDCS + CRP + M | prefrontal | 1.1 mA, 20 min, 10 times/w | 14/12 | 26/0 | 64.2±5.5 | - | 19.7±2.6 | 4 w |
| | | no tDCS + CRP + M | | | 16/10 | 26/0 | 62.2±6.5 | - | 19.5±2.6 | |
| Zheng 2017 | RCT | anodal tDCS + CRP + M | affected frontotemporal | 0.5 mA, 20 min, 5 times/w | 18/9 | 27/0 | 63.31±6.72 | 24.49±10.78 d | - | 6 w |
| | | no tDCS + CRP + M | | | 17/11 | 28/0 | 62.11±6.82 | 25.45±9.51 d | - | |

*(Continued)*

**Table 1.** (Continued)

| Study | Design | Intervention | | | Male/Female | Ischemic/Hemorrhage | Ages | Onset | MMSE | Follow-up |
|---|---|---|---|---|---|---|---|---|---|---|
| | | Treatments | Location | Intensity | | | | | | |
| Wang 2018 | RCT | anodal tDCS + CRP | - | 1.0 mA, 20 min, 5 times/w | 36/7 | - | 62.1±5.8 | - | - | 8 w |
| | | no tDCS + CRP | - | | 32/8 | - | 64.2±4.9 | - | - | |
| Hosseinzadeh 2018 | RCT | anodal tDCS + CRP | superior-temporal | 2.0 mA, 10 min, 3 times/w | 12/13 | 25/0 | 58±8 | - | - | 4 w |
| | | sham tDCS + CRP | | | 12/13 | 25/0 | 59±7 | - | - | |

RCT, randomized controlled trial; tDCS, transcranial direct current stimulation; CRP, traditional rehabilitation program; M, medications; h, hours; d, day; w, week; m, month.

sham tDCS or no tDCS was considered as passive stimulus.

"-", not applicable or nothing to note;

"*", the study contains two independent trials.

## Risk of bias within the studies

The risk of bias judgments for each trial is displayed in S3 Table. The overall risk of bias ranged from moderate to critical among the included studies; 8 (53.3%) reported their randomization sequence generation, while only 1 (6.7%) concealed the allocation, which may have caused a selection bias. Only 5 (33%) studies used sham tDCS as a placebo condition, and 4 (27%) studies blinded outcome assessment, which may have resulted in performance and detection biases, respectively. uncertainty about the reporting bias and attrition bias does exist since no priori published trial protocols for the included studies were found.

## Synthesis of results

Thirteen studies involving 692 participants were included in the meta-analyses. In addition to comprehensive testing, including the Minimum Mental State Examination and Montreal Cognitive Assessment, we also analyzed memory and attention performance specifically. Subgroup analyses according to the types of stroke, comparators, poststroke follow-up duration were presented together with their overall effects.

**General cognitive function.** Eleven RCTs involving 582 patients were included in this outcome analysis; 292 participants were treated with anodal tDCS, and 290 were in the control groups. The random-effects model was applied because of the significant heterogeneity among the studies ($I^2 = 78\%$, $P < 0.01$). We found that, compared with sham tDCS or no tDCS intervention, patients in the anodal tDCS group were more likely to have higher points on the cognitive function test (SMD = 1.31, 95% CI 0.91–1.71, $P < 0.00001$, $I^2 = 78\%$) at the end of follow-up (Fig 2). Sensitivity analysis showed that Song (2019) is a major source of heterogeneity and they did not report the duration of treatment. After removing this study, the SMD for cognitive function test was 1.17 (95% CI 0.83–1.51, $P < 0.00001$, $I^2 = 78\%$). For this reason, we omitted this study from the subgroup analyses.

The result of subgroup analysis showed that there was no statistically significant difference between sham tDCS and no tDCS groups, with pooled SMDs of 0.81 (95% CI 0.27–1.35, $P < 0.00001$, $I^2 = 0\%$) and 1.24 (95% CI 0.86–1.63, $P < 0.00001$, $I^2 = 74\%$), respectively. The test of subgroup difference gave $X^2 = 1.60$ and $P = 0.21$ (Fig 3). However, subgroup analysis showed the pooled SMDs of ischemic and hemorrhagic stroke were 1.27 (95% CI 0.89–1.64, $P < 0.00001$, $I^2 = 49\%$) and 0.80 (95% CI 0.45–1.15, $P < 0.00001$, $I^2 = 30\%$), respectively, the

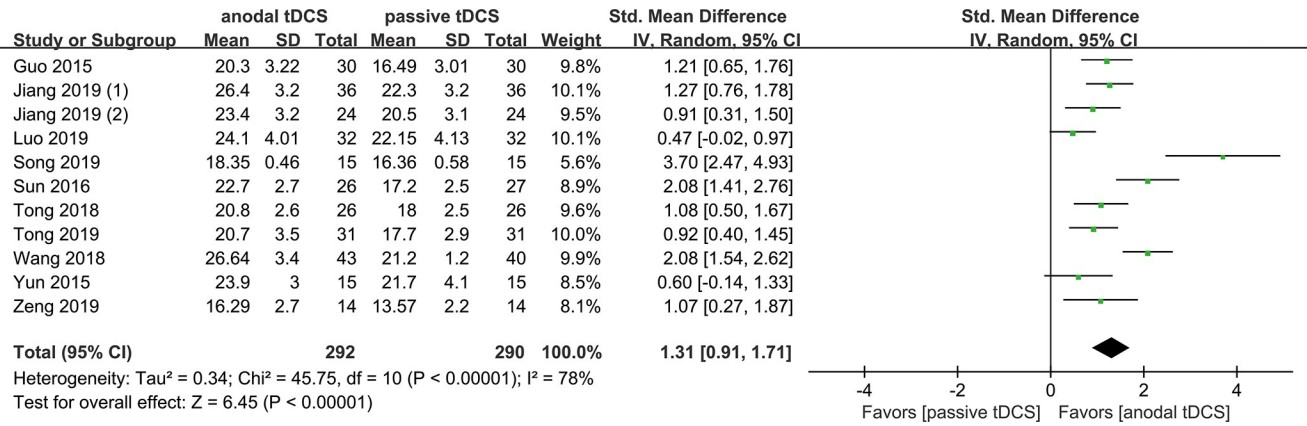

**Fig 2. Forest plot of general cognitive function assessed by MMSE or MoCA.**

test of subgroup difference gave $X^2 = 3.11$ and $P = 0.08$ (Fig 4). Similarly, in a third subgroup analysis, we found that the effects of tDCS was associated with the onset of stoke, with pooled SMDs of 1.34 (95% CI 0.90–1.78, $P < 0.00001$, $I^2 = 57\%$) for < 40 days and 0.72 (95% CI 0.42–1.03, $P < 0.00001$, $I^2 = 0\%$) for ≥ 40 days, the test of subgroup difference gave $X^2 = 5.11$ and $P = 0.02$ (Fig 5).

**Attention performance.** Six studies involving 286 patients separately tested the level of attention or concentration at the end of the intervention period (Fig 6 [25–28, 33, 37]). The pooled analysis indicated that anodal tDCS can further improve attention or concentration performance (SMD = 0.66, 95% CI 0.11–1.20, $P = 0.02$, $I^2 = 79\%$).

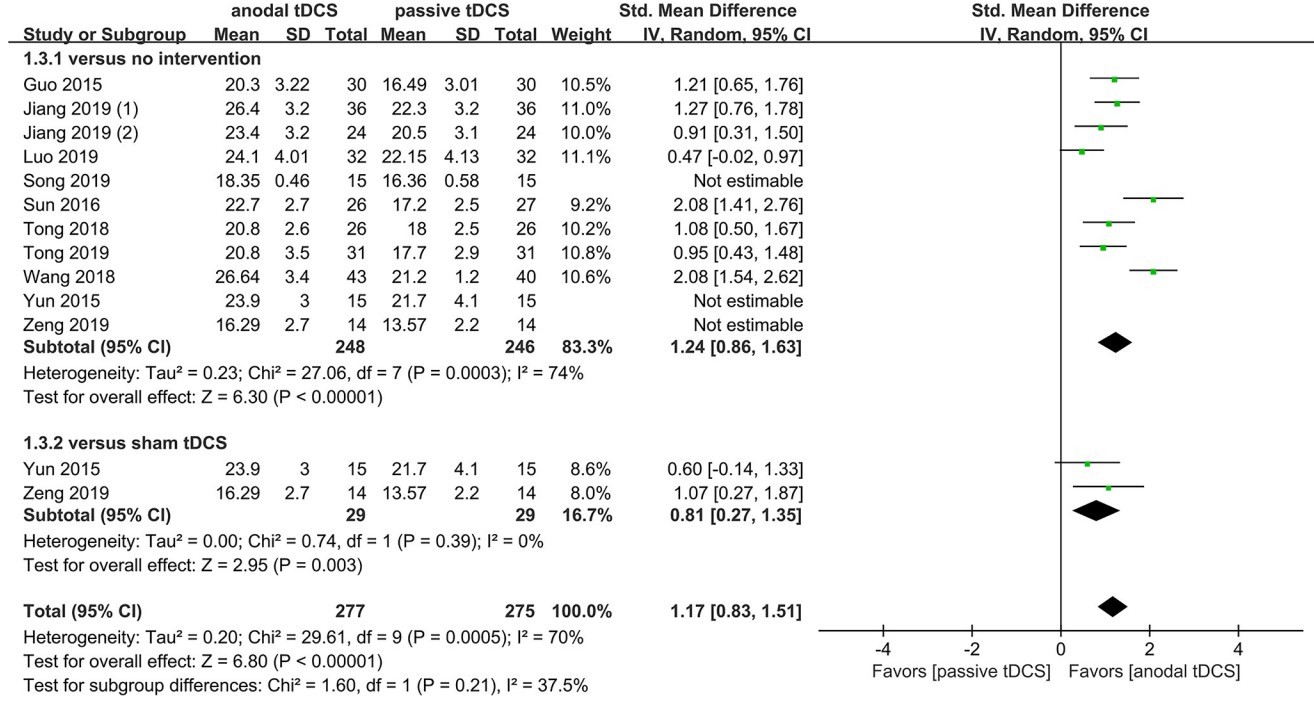

**Fig 3. Subgroup analysis based on different comparators.**

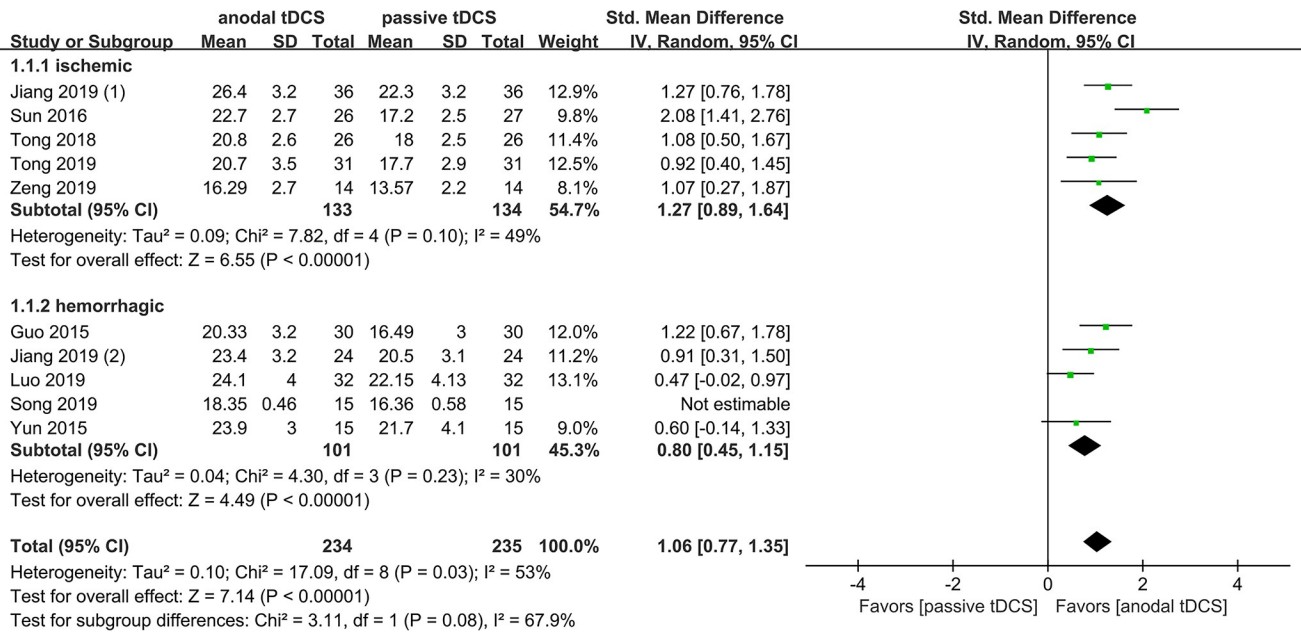

**Fig 4. Subgroup analysis based on different type of stroke.**

**Memory performance.** Three studies with a total of 150 participants examined memory function by different methods (Fig 7 [25, 26, 28]). We found no evidence of an effect of tDCS on memory function when we analyzed the data in comparison with passive tDCS groups (SMD = 0.41, 95% CI -0.67–1.50, P = 0.46). The random-effects model was used due to the significant heterogeneity among the studies ($I^2$ = 89%, P = 0.0001).

In addition, two other studies excluded from the statistical pooling both reported evidence of effects in favor of anodal tDCS regarding measures of cognitive function.

**Publication bias.** Publication bias seemed to be unlikely according to the inspection of the funnel plots for studies examining the effect of anodal tDCS versus passive tDCS on

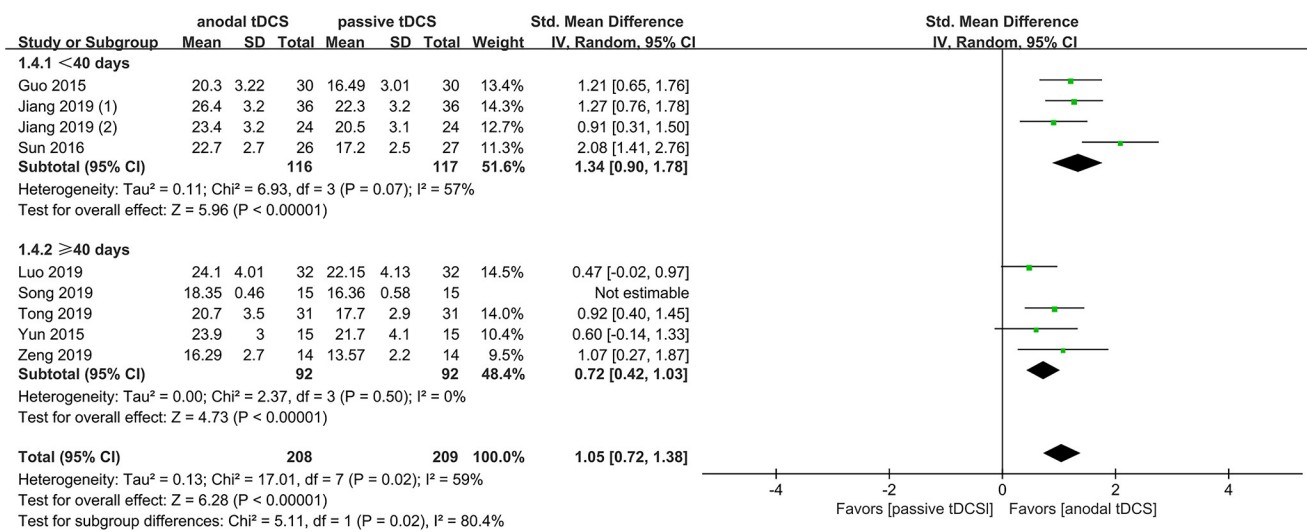

**Fig 5. Subgroup analysis based on stroke duration.**

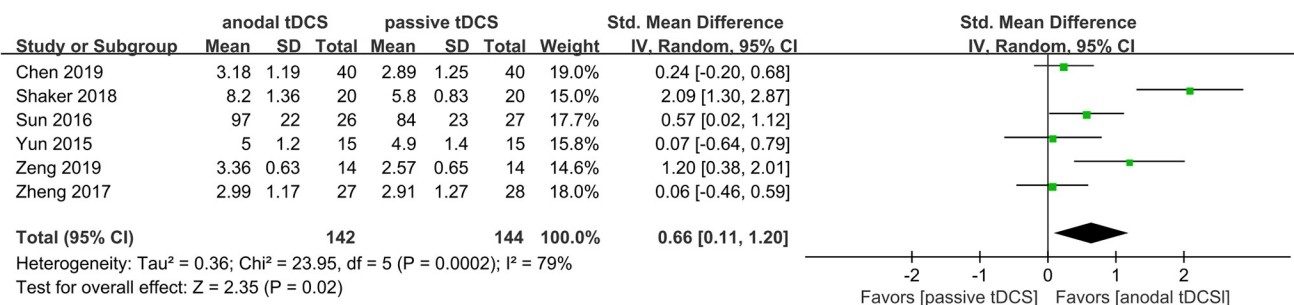

**Fig 6. Forest plot of attention performance.**

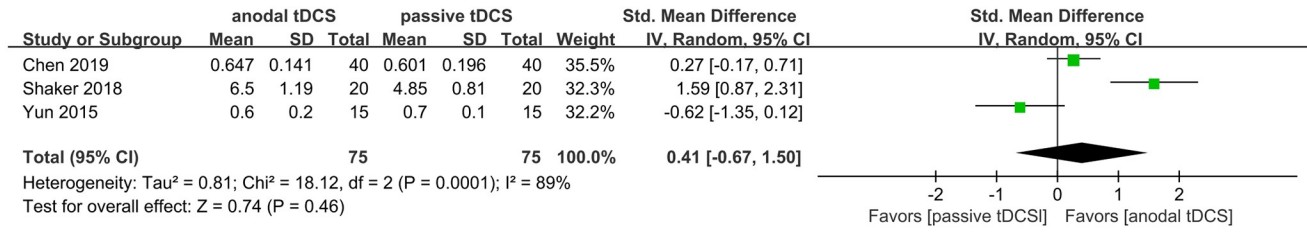

**Fig 7. Forest plot of memory performance.**

general cognition (Fig 8); additionally, the Egger test ($P = 0.13$) detected no significant small-study effects.

## Discussion

### Summary of evidence

This meta-analysis of 13 studies showed that anodal tDCS was associated with improved general cognitive performance in stroke patients. According to the subgroups analyses, the duration of poststroke and type of stroke were found to have a significant impact on the effects of tDCS. Furthermore, anodal tDCS was beneficial for attention specifically but not for memory specifically.

### Comparison with other studies

To the best of our knowledge, this is the first meta-analysis to evaluate the effect of anodal tDCS on cognitive recovery after stroke. One earlier systematic review discussed the effects of tDCS on activities of daily living and physical and cognitive functioning after stroke, but only one RCT involving cognition was included; thus, no statistical pooling was performed in this field [39]. In fact, the current findings on the effects of tDCS for other dysfunctions after stroke are inconsistent. Several systematic reviews and meta-analyses have shown that the tDCS is beneficial for poststroke motor function and aphasia [10, 40, 41], whereas others have reported that the effects of tDCS are similar to those of sham treatment [11, 42]; one study indicated that tDCS did not improve gait and ambulation performance poststroke [43]. Moreover, in 2017, the evidence-based tDCS guidelines made no recommendations for motor function or aphasia because the level of evidence was not sufficient to ensure efficacy [7].

### Strengths and limitations

The strengths of the present systematic review lie in the comprehensive literature retrieval and quantitative synthesis: we conducted an extensive search of the Chinese database, and we

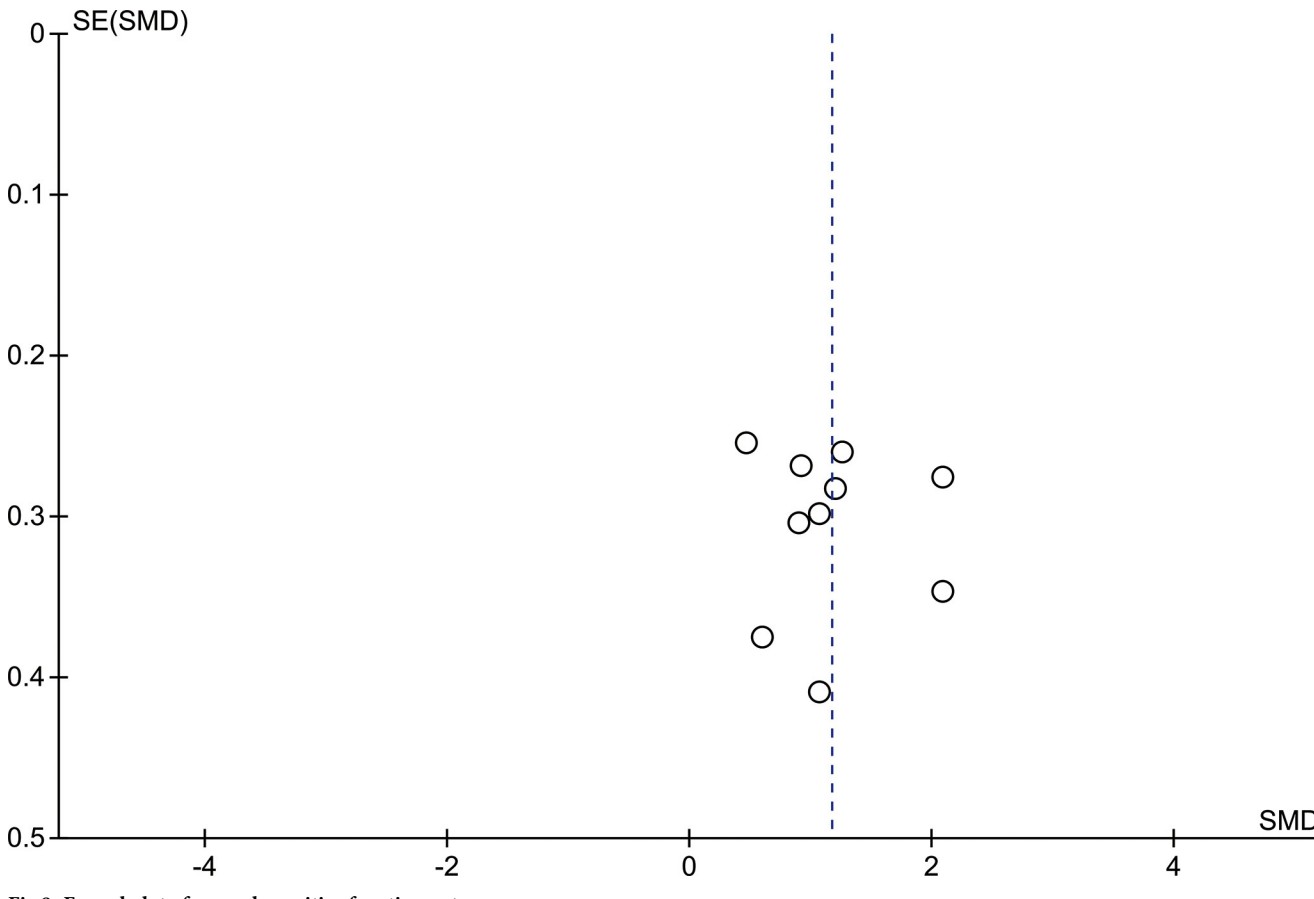

**Fig 8. Funnel plot of general cognitive function outcome.**

obtained several recently published RCTs; we strictly followed the inclusion criteria; multiple subgroup analyses were performed. However, our study still has several limitations. First, 73% of the included studies were from China and there was a lack of English studies. Second, the majority of the included studies had lower methodological quality since they did not report the details of the randomization sequence generation or allocation concealment; there was a lack of sham tDCS in the control group; uncertainty about the reporting bias and attrition bias does exist since no priori published trial protocols for the included studies were found. Third, the included studies were different in terms of population (age, lesion site, levels of impairment) and stimulation methodologies, so our results were derived from heterogeneous data. Fourth, safety-relevant indicators were not presented in this meta-analysis due to the lack of data.

## Implications for practice

This meta-analysis suggests that stroke patients with cognitive function deficits appear to benefit from transcranial direct current stimulation. However, considering the risk of bias of included studies, tDCS cannot yet be recommended as a standard therapy for the stokes. More research is needed to determine the potential benefits of tDCS in the future.

## Implications for research

Although the results of this meta-analysis support the conclusion that the anodal tDCS proto- col showed promising effects on cognition poststroke, we were unable to determine the

effectiveness of anodal tDCS in cognitive progression due to the heterogeneity of the participant characteristics, stimulation methods and methodological deficiencies of the included studies.

Based on the current findings, the tDCS appears to be more effective in patients with shorter course of disease and ischemic stroke. But, it remains unclear whether the stimulation parameters associated with the efficacy of anodal tDCS. It's worth noting that there was wide variation in the tDCS stimulation parameters including intensity, the number of sessions and duration as well as electrode location; all of these parameters are moderators of the cumulative effect, which may in fact be more relevant to the outcome. In addition, the levels of impairment, the affected area, the recovery stage may also affect the outcome of tDCS. Conducting more high quality and large sample studies for both conditions may further reveal these influencing factors. It would be necessary to use sham tDCS as a placebo, as it seemed to reduce the possibility of exaggerating the tDCS effects.

Although current studies have shown that tDCS may be effective on cognitive function recovery after stroke, the number of studies focusing on specific cognitive functions is limited. Cognitive impairment manifests itself in many ways, and these manifestations may be related to one another [44]. It is unclear whether tDCS plays a different role in different domains of cognition. Thus, we suggest that studies should focus on each specific domain, in addition to general cognition measurements.

## Conclusions

The meta-analysis suggests that anodal tDCS might improve poststroke cognition as examined by two generally used methods: MMSE and MoCA. Significant improvement was also found when attention performance was analyzed separately, while the results showed no difference in memory performance. However, the evidence on the impact of patient characteristics and stimulation parameters is still lacking. More high-quality research is needed to determine the efficacy of tDCS in the treatment of cognitive deficits after stroke and to establish the optimal treatment program.

## Supporting information

**S1 Table. The preferred reporting items for systematic reviews and meta-analyses (PRISMA) 2009 checklist.**
(PDF)

**S2 Table. Search strategies.**
(PDF)

**S3 Table. Risk of bias for the included studies as judged by the cochrane collaboration risk of bias tool.**
(PDF)

## Author Contributions

**Conceptualization:** Ru-bing Yan.

**Data curation:** Ru-bing Yan, Xiao-li Zhang, Jing-ming Hou.

**Formal analysis:** Ru-bing Yan, Xiao-li Zhang.

**Funding acquisition:** Hong-liang Liu.

**Investigation:** Xiao-li Zhang.

**Methodology:** Xiao-li Zhang, Yong-hong Li.

**Project administration:** Hong-liang Liu.

**Resources:** Hong-liang Liu.

**Writing – original draft:** Ru-bing Yan.

**Writing – review & editing:** Yong-hong Li, Han Chen.

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
