## [Decision Letter · Decision Letter 0]

19 Nov 2019

PONE-D-19-28981

Effect of transcranial direct current stimulation on cognitive function in stroke patients: a systematic review and meta-analysis

PLOS ONE

Dear Dr. Liu,

Thank you for submitting your manuscript to PLOS ONE. Your paper was reviewed by two experts on the brain stimulation field, both of them found that the aim of the study has high relevance, however, they have also indicated that the present version of the paper suffers from many shortcomings, mainly related to the interpretation of the results. After careful consideration, we feel that it has merit but does not fully meet PLOS ONE’s publication criteria as it currently stands. Therefore, we invite you to submit a revised version of the manuscript that addresses the all points raised during the review process.

We would appreciate receiving your revised manuscript by 19th of January, 2020. To enhance the reproducibility of your results, we recommend that if applicable you deposit your laboratory protocols in protocols.io, where a protocol can be assigned its own identifier (DOI) such that it can be cited independently in the future. For instructions see: http://journals.plos.org/plosone/s/submission-guidelines#loc-laboratory-protocols

We look forward to receiving your revised manuscript.

Kind regards,

Andrea Antal, PhD

Academic Editor

PLOS ONE

Journal Requirements:

Reviewers' comments:

Reviewer's Responses to Questions

**Comments to the Author**

1. Is the manuscript technically sound, and do the data support the conclusions?

Reviewer #1: No

Reviewer #2: Partly

2. Has the statistical analysis been performed appropriately and rigorously? 

Reviewer #1: No

Reviewer #2: Yes

3. Have the authors made all data underlying the findings in their manuscript fully available?

Reviewer #1: Yes

Reviewer #2: Yes

4. Is the manuscript presented in an intelligible fashion and written in standard English?

Reviewer #1: No

Reviewer #2: No

5. Review Comments to the Author

Reviewer #1: General assessment:

The article fits the scope of the journal. The research is novel. The title is representative of the articles contents. The abstract summarises the contents clearly. The state-of-the-art is well described in the knowledge gap defined. The objectives are well articulated. The applied research methodology is solid, but the comparator as a covariate in the included studies should be taken into account to a greater extent. The review has been registered retrospectively. The results are reliable, but of limited validity and the objectives have partly been reached. Important limitations arising from the choice of comparators should correctly be mentioned. The conclusions are at the moment not justified.

Detailed assessment (the pagination refers to the PDF-preprint):

your work examines the effects of anodal tDCS on cognition after stroke. To make this clear to the reader, I suggest to every time referred to „anodal tDCS“ throughout the manuscript, where appropriate. I would also recommend to check for language errors once again.

Page nine: the Prospero-ID presented in the abstract is wrong and refers to another publication. Please correct.

Page 9: you state, that tDCS was firstly developed in 1992. Please provide the corresponding citation for that.

Page 11: the term used for describing the comparator „passive stimulus“ seems to be a bit ambiguous, here. Apparently, you subsumed no additional intervention and sham tDCS under this term (as can be seen from table 1) and I would suggest to describe this here, also.

Page 12: please check the last sentence of the last paragraph in „outcome measures“, there seems to be a word missing.

Page 13: please cite the computer program review manager 5.3, as requested in the programme.

Page 13: „The publication bias was unable to be assessed due to a small number of studies included.“ I would regard this sentence as a part of the discussion section and hereby recommend to move it there.

Page 13: there seems to be a typo: „we determined that 15 studies with 16 trails“, shouldn‘t it be „trials“ here?

Figure 1: you excluded 33 studies, but the reasons are only given for 15. Moreover, you screened 49 full text articles and excluded 33 studies and this resulted in 15 included studies, but 49-33 should be 16. Please clarify. If the difference is due to an inequality of the number of fulltext publications, I would recommend to explain this to the reader with a comment in the figure caption.

Page 14: in the last sentence, after CNT and LOTCA, are these signs, commata? Please check.

Table 1: in Park 2013 there seems to be a typo (CPR instead of CRP).

Page 17: in the risk of bias section, you stated all trials reported complete outcome data and there were no selective reporting among the studies. I am a bit curious, how you did proof for complete report of outcome data, has there been an a priori published trial protocol for every single included study and you have checked it and there was no deviation? If no, I would disagree with your risk of bias rating and personally would rate it at least as unclear.

Table S3: blinding of participants and personnel - there are only five studies, who used sham tDCS as an adequate method of blinding of participants. I would disagree with your risk of bias rating in the studies which did not use sham tDCS and not rated as high risk of bias in this regard, since this might have introduced superior performance bias because of the absence of a placebo effect. Please check.

Page 11, Analysis: in order to reduce bias, introduced by analysing studies with different comparators (sham tDCS versus no sham tDCS) I would recommend to restructure the results section as follows (after having a look at the interventions listed in table 1):

analysis one: active tDCS versus sham tDCS

analysis tool: active tDCS versus no intervention

for reasons of simplicity, you may want to combine results MMSE and MoCA, if you regard both outcome measures measuring the same outcome.

Page 11: in the analysis corresponding to figure 2, you have calculated SMD on the basis of change scores (Park 2013) and total values. However, there is is mathematically not allowed. You can have a look into the Cochrane Handbook for further instruction.

Figure 3: in this subgroup analysis, the study „Tong 2018“ suddenly has got a different sample size and different means and standard deviations when compared to the analyses of figure 2, but shouldn‘t it be the same question please check.

Page 18: „Specialized in attention function“ I would recommend to rename this outcome into „attention“ for „attention function“ or something similar, since otherwise it may be hard for the reader to understand meaning.

Page 19: in the discussion section, you interpret the SMDs of your analyses in units of the underlying outcome measure „The improvement was around 1.01 points for the MMSE and 3.18 points for the MoCA.“ This is wrong. Please interpreted it in units of standard deviations or back transform it to the original outcome measure (the Cochrane Handbook gives advice on this).

Page 20: last sentence of comparing with others studies - but there is not a univocal corpus of evidence, that tDCS is beneficial… For a comprehensive overview of current reviews on this topic, you may have a look into the published Cochrane reviews of this or a related topic or into the tDCS guideline of Lefaucheur et al. (2017).

Page 20: strengths and limitations: I would add here the threat publication of bias and the comparator issue (sham tDCS versus no intervention), if appropriate.

Page 21, implications for research: you state, that „intensity (≥1.8 Ma or ≤1.5 Ma)“, but shouldn‘t it be „intensity (≥1.5 mA or ≤1.5 mA)“ instead? Please check.

Overall, a well written review with some methodological flaws in an emerging topic with very recent evidence.

Reviewer #2: In the paper, the authors aim to present a systematic review of the effects of transcranial stimulation on cognitive function in stroke patients. They also provide a meta analysis combining results from several reports that have evaluated the effects of this therapeutic technique after stroke; specifically, in cognitive function.

As the authors state, the effects of transcranial stimulation after stroke have been mainly focused on motor or speech ability, however, many of these patients have cognitive deficits that can represent important impairments in their quality of life. Rigorous systematic reviews about the effectiveness of tDCS in healthy individuals regarding cognitive function, have shown little-to-no reliable neurophysiologic and cognitive effects (Horvath et al., 2015). However, some researchers argue that the effects of tDCS in cognitive function might be more evident after cognitive disfunction. Therefore, evaluating if transcranial stimulation results in cognitive improvements after stroke, could be of significant value, considering the low cost and invasiveness of transcranial stimulation.

The review is generally sound in its methodology by having two reviewers that independently screened the studies, extracted the data, and evaluated the quality of included studies using the Cochrane Collaboration Risk of Bias Tool. I consider, that as a generality the inclusion criteria is adequate. However, when measuring the effects of tDCS after stroke, many other criteria comes in to play, making it difficult to assess the real effectiveness of the technique. For example, the study does not consider dividing results by the affected area, the extent of infarct, and the post-stroke time-line from treatment. They do present some information about such factors, but it needs a much stronger discussion and clarification on how these differences in population and methodologies may affect the results.

In general, I consider that although the study is methodologically valid, the language is unclear, making it difficult to follow. The authors should revise the language to improve readability. Additionally, I consider that there needs to be a more critical discussion of the results in order to consider the real state of the contribution of tDCS to cognitive function after stroke. Taking into consideration that this is a systematic review, the discussion is not very clear or needs further and more critical explorations. I think this could be resolved only with structural changes to the manuscript and noting that the availability of studies (due to variability of methods and populations) can not derive a reliable conclusion on the effects of tDCS in cognitive processes. The evidence shown by the authors, does point to some cognitive improvements, but there might be many considerations before arriving to a strong conclusion; the paper needs to have such discussion.

Some specific points of improvement are the following:

1. The reference in the following sentence does not reflect the information contained in authors’ statements that: “Stroke ranks No. 5 among all causes of death and is a leading cause of long-term disability in the world, which produces a major burden to society [1].”

[1] Burke JF, Lisabeth LD, Brown DL, Reeves MJ, Morgenstern LB. Determining stroke's rank as a

cause of death using multicause mortality data. Stroke. 2012;43(8):2207-11.

2. The authors claim that: “As a neuromodulatory approach, tDCS works by depolarizing or hyperpolarizing neuronal membrane potentials through the activation of sodium- and calcium-dependent channels and NMDA receptor activity, thereby modulating neuronal excitability”. However, no references are provided to support the claim.

3. Authors state “…and some preliminary studies also shown beneficial effects of tDCS on cognitive functions both in healthy subjects and stroke patients [9,10]”. However, authors cite:

Webster B, Celnik P, Cohen L. Noninvasive Brain Stimulation in Stroke Rehabilitation.

Fregni F, Boggio PS, Mansur CG, Wagner T, Ferreira MJL, Lima MC, et al. Transcranial direct current stimulation of the…

Please cite specific effects and studies.

4. On the results section the authors state that: “The results of subgroup analysis suggested that the effectiveness of tDCS seems to be slightly better for ischemic stroke than hemorrhagic stroke, but there was not significant difference.” However, in their conclusions they state: "tDCS is likely to be effective for patients with cognitive impairment after stroke, and the effect might be different among hemorrhagic and ischemic stroke". This conclusion doesn’t seem to be based on facts, at least from their own results.

5. Please consider revising the language specifically of the following paragraphs, in order to be able to follow the study properly:

“Except for high morbidity and mortality, the burden of stroke related-disability is another big problem in survivors, of which, the incidence of poststroke cognitive impairment (PSCI) is ranging from 22% to 47% in different studies [3-5], it has had a serious impact on both the economic and the quality of life.”

“Transcranial direct current stimulation (tDCS) was firstly developed in 1992 for

clinical purposes; nowadays, tDCS constitutes as a promising method in neurological

condition regulating.”

“Specifically, some other measurements are used for the evaluation of

attention and memory ability separately, including Computerized Neuropsychological

Test (CNT), Loewenstein occupational therapy cognitive assessment (LOTCA) and

various other forms of testing used in clinical.”

“Seven trials included two stroke types, seven studies focused on ischemic stroke and one studies on hemorrhagic stroke alone.”

6. PLOS authors have the option to publish the peer review history of their article (what does this mean?). If published, this will include your full peer review and any attached files.

Reviewer #1: Yes: Bernhard Elsner

Reviewer #2: No

---

## [Author Response · Author response to Decision Letter 0]

15 Jan 2020

Dear Editor and Reviewers:

We are truly grateful to yours critical comments and thoughtful suggestions concerning our manuscript entitled “Effect of transcranial direct current stimulation on cognitive function in stroke patients: a systematic review and meta-analysis” (ID: PONE-D-19-28981). We feel lucky that our manuscript went to these reviewers as the valuable comments from them really helped us with the improvement of our manuscript. Based on the comments we received, careful modifications have been made to original manuscript. All changes were marked in red text. In addition, we have consulted native English speakers for paper revision before the submission this time. We hope the new manuscript will meet your magazine’s standard. Below you will find our point-by-point responses to the reviewers’ comments/ questions:

Reviewer #1:

1. Response to comment: your work examines the effects of anodal tDCS on cognition after stroke. To make this clear to the reader, I suggest to every time referred to “anodal tDCS“ throughout the manuscript, where appropriate. I would also recommend to check for language errors once again.

Response: it is really true as reviewer suggested that we should every time referred to the same word throughout the manuscript. However, during the process of revising, we realized that the “active tDCS” are more suitable than “anodal tDCS”, so we have replaced it and also checked the language errors again.

2. Response to comment: page nine: the Prospero-ID presented in the abstract is wrong and refers to another publication. Please correct.

Response: we are very sorry for our incorrect writing of Prospero-ID, after checking, we have corrected the Prospero-ID as: CRD 42019137191.

3. Response to comment: page 9: you state, that tDCS was firstly developed in 1992. Please provide the corresponding citation for that.

Response: we are very sorry for our negligence of citing the reference here and we have added a new one now.

4. Response to comment: page 11: the term used for describing the comparator “passive stimulus” seems to be a bit ambiguous, here. Apparently, you subsumed no additional intervention and sham tDCS under this term (as can be seen from table 1) and I would suggest to describe this here, also.

Response: as reviewer suggested, we should describe the “passive stimulus” more clearly. At this point, we have also realized that the “anodal tDCS” can also lead to ambiguity of using only anodal electrodes without cathodal ones. Hence, in the revised manuscript, we have replaced the “anodal tDCS” with “active tDCS”, and made a explanation for “passive tDCS” in main text as well as table 1. 

5. Response to comment: page 12: please check the last sentence of the last paragraph in “outcome measures“, there seems to be a word missing.

Response: considering the reviewer’s suggestion, we have re-written this sentence. 

6. Response to comment: page 13: please cite the computer program review manager 5.3, as requested in the programme.

Response: as suggested by the reviewer, we have provided a direct link to the review manager 5.3 within the paper.

7. Response to comment: page 13: “The publication bias was unable to be assessed due to a small number of studies included.” I would regard this sentence as a part of the discussion section and hereby recommend to move it there.

Response: it is really true as reviewer suggested that this sentence should be part of the discussion section. In addition, we found that it is possible and necessary to evaluate publication bias after changing the method of results synthesis(combining the outcomes of MMSE and MoCA). Therefore, we have added the method of publication bias analysis in the subgroup analysis section and showed the result of publication bias in the result section. 

8. Response to comment: page 13: there seems to be a typo: “we determined that 15 studies with 16 trails“, shouldn‘t it be “trials“ here?

Response: we are very sorry for our incorrect writing of “trials”, we’ve corrected it.

9. Response to comment: figure 1: you excluded 33 studies, but the reasons are only given for 15. Moreover, you screened 49 full text articles and excluded 33 studies and this resulted in 15 included studies, but 49-33 should be 16. Please clarify. If the difference is due to an inequality of the number of full text publications, I would recommend to explain this to the reader with a comment in the figure caption.

Response: we are so sorry for our careless and mistakes in Figure 1. The reasons for excluded studies are not shown completely due to the picture display problem, we have modified the picture. Besides, we missed one excluded study, so we actually excluded 34, leaving 15. Additionally, after this revision, we have removed another study with unavailable data form meta-analysis, finally, 15 studies were included and 13 studies were involved in quantitative synthesis.

10. Response to comment: page 14: in the last sentence, after CNT and LOTCA, are these signs, commata? Please check.

Response: thank you very much for bringing this to our attention, we carefully reviewed the sentence and decided that it may be inappropriate to place it here, so we have finally deleted this sentence.

11. Response to comment: Table 1: in Park 2013 there seems to be a typo (CPR instead of CRP).

Response: we apologize for the error of the “CRP” in Table 1 and have corrected. 

12. Response to comment: page 17: in the risk of bias section, you stated all trials reported complete outcome data and there were no selective reporting among the studies. I am a bit curious, how you did proof for complete report of outcome data, has there been an a priori published trial protocol for every single included study and you have checked it and there was no deviation? If no, I would disagree with your risk of bias rating and personally would rate it at least as unclear.

Response: thank you very much for pointing out the problems in “risk of bias” rating, and we fully accept the reviewer's suggestions. Therefore, we have reassessed the risk of bias: since no priori trial protocol was found, we rated the domains of complete outcome data and selective reporting as unclear.

13. Response to comment: Table S3: blinding of participants and personnel - there are only five studies, who used sham tDCS as an adequate method of blinding of participants. I would disagree with your risk of bias rating in the studies which did not use sham tDCS and not rated as high risk of bias in this regard, since this might have introduced superior performance bias because of the absence of a placebo effect. Please check.

Response: thank you so much for bringing this to our attention. It is really true as reviewer suggested that the absence of a placebo effect for control groups is easily lead to performance bias. So we have corrected the results in “risk of bias” we have rated these studies without sham tDCS as high risk of bias in terms of blinding of participants and personnel. 

14. Response to comment: Page 11, Analysis: in order to reduce bias, introduced by analyzing studies with different comparators (sham tDCS versus no sham tDCS) I would recommend to restructure the results section as follows (after having a look at the interventions listed in table 1):

analysis one: active tDCS versus sham tDCS

analysis tool: active tDCS versus no intervention

for reasons of simplicity, you may want to combine results MMSE and MoCA, if you regard both outcome measures measuring the same outcome.

Response: thank you very much for these valuable suggestions. According to the reviewer’s suggestion, we have combined the results of MMSE and MoCA and changed the way the results were synthesized, since both of them measure the general cognitive status on a scale of 30 points. Hence, the part of the result has changed quite a bit. In this case, we have performed more subgroup analyses including those based on different comparators (sham tDCS versus no tDCS).

15. Response to comment: page 11: in the analysis corresponding to figure 2, you have calculated SMD on the basis of change scores (Park 2013) and total values. However, there is mathematically not allowed. You can have a look into the Cochrane Handbook for further instruction.

Response: we are very sorry for our incorrect selection of summary statistics. As reviewer suggested, we looked at the Cochrane Handbook, found that final value and change scores should not be combined together as SMD. After the discussion, we’ve excluded park 2013 from this meta-analysis because no total values was reported in this study.

16. Response to comment: Figure 3: in this subgroup analysis, the study “Tong 2018“ suddenly has got a different sample size and different means and standard deviations when compared to the analyses of figure 2, but shouldn‘t it be the same question please check.

Response: we are very sorry for our incorrect writing of the sample size, means and standard deviations in original Figure 3. Based on these comments and suggestions, we have made a lot of changes to the result section and also reedited Fig 2 and Fig 3.

17. Response to comment: Page 18: “Specialized in attention function“ I would recommend to rename this outcome into “attention“ for “attention function“ or something similar, since otherwise it may be hard for the reader to understand meaning.

Response: thank you so much for bringing this to our attention. We have rename this outcome as “Attention performance” according to the reviewer’s suggestion.

18. Response to comment: page 19: in the discussion section, you interpret the SMDs of your analyses in units of the underlying outcome measure “The improvement was around 1.01 points for the MMSE and 3.18 points for the MoCA.“ This is wrong. Please interpreted it in units of standard deviations or back transform it to the original outcome measure (the Cochrane Handbook gives advice on this).

Response: we are very sorry for our incorrect interpreting of SMDs as a outcome measure. Based on these comments and suggestions, we have carefully revised the results section and re-selected “MD” as outcome measure.

19. Response to comment: page 20: last sentence of comparing with others studies - but there is not a univocal corpus of evidence, that tDCS is beneficial… For a comprehensive overview of current reviews on this topic, you may have a look into the published Cochrane reviews of this or a related topic or into the tDCS guideline of Lefaucheur et al. (2017).

Response: thank you very much for providing us with these valuable studies. After careful reading, we have re-written the “Comparing with other studies” part according to your suggestions. 

20. Response to comment: page 20: strengths and limitations: I would add here the threat publication of bias and the comparator issue (sham tDCS versus no intervention), if appropriate.

Response: we have made correction according to the reviewer’s comments. Since we have combined the outcome measures of MMSE and MoCA, we added the analysis of publication bias in the revised manuscript. And we have also performed a subgroup analysis based on different comparators (sham tDCS versus no intervention).

21. Response to comment: page 21, implications for research: you state, that “intensity (≥1.8 Ma or ≤1.5 Ma)”, but shouldn’t‘t it be “intensity (≥1.5 mA or ≤1.5 mA)” instead? Please check.

Response: we are very sorry for our mistake here. The sentence has been deleted because of structural changes to the discussion section. 

Reviewer #2:

1. Response to comment: the reference in the following sentence does not reflect the information contained in authors’ statements that: “Stroke ranks No. 5 among all causes of death and is a leading cause of long-term disability in the world, which produces a major burden to society [1].”

[1] Burke JF, Lisabeth LD, Brown DL, Reeves MJ, Morgenstern LB. Determining stroke's rank as a

cause of death using multicause mortality data. Stroke. 2012;43(8):2207-11.

Response: thanks very much for your careful checking of our manuscript. We have revisited the reference and re-written this sentence.

2. Response to comment: the authors claim that: “As a neuromodulatory approach, tDCS works by depolarizing or hyperpolarizing neuronal membrane potentials through the activation of sodium- and calcium-dependent channels and NMDA receptor activity, thereby modulating neuronal excitability”. However, no references are provided to support the claim.

Response: we are very sorry for our negligence of providing the references here. In terms of the mechanics of tDCS, we referred to these studies:

(1).Lefaucheur JP, Antal A, Ayache SS, Benninger DH, Brunelin J, Cogiamanian F, et al. Evidence-based guidelines on the therapeutic use of transcranial direct current stimulation (tDCS). Clin Neurophysiol. 2017;128(1):56-92.

(2).Paulus W. Transcranial direct current stimulation (tDCS). Suppl Clin Neurophysiol. 2003;56:249-54.

And we have now provided these references in the revised version.

3. Response to comment: authors state “…and some preliminary studies also shown beneficial effects of tDCS on cognitive functions both in healthy subjects and stroke patients [9,10]”. However, authors cite:

Webster B, Celnik P, Cohen L. Noninvasive Brain Stimulation in Stroke Rehabilitation.

Fregni F, Boggio PS, Mansur CG, Wagner T, Ferreira MJL, Lima MC, et al. Transcranial direct current stimulation of the…

Please cite specific effects and studies.

Response: it is really true as reviewer suggested that we should cite more specific studies here. therefore, we have deleted the original citations and added new ones that focus on the effects of tDCS for healthy or stroke participants.

Additional references are: 

(1). Hsu WY, Ku Y, Zanto TP, Gazzaley A. Effects of noninvasive brain stimulation on cognitive 

(2). function in healthy aging and Alzheimer's disease: a systematic review and meta-analysis. NEUROBIOL AGING. 2015;36(8):2348-59.

(3). Clemens B, Jung S, Zvyagintsev M, Domahs F, Willmes K. Modulating arithmetic fact retrieval: a single-blind, sham-controlled tDCS study with repeated fMRI measurements. NEUROPSYCHOLOGIA. 2013;51(7):1279-86.

(4). Jo JM, Kim YH, Ko MH, Ohn SH, Joen B, Lee KH. Enhancing the working memory of stroke patients using tDCS. Am J Phys Med Rehabil. 2009;88(5):404-9.

(5). Kazuta T, Takeda K, Osu R, Tanaka S, Oishi A, Kondo K, et al. Transcranial Direct Current Stimulation Improves Audioverbal Memory in Stroke Patients. Am J Phys Med Rehabil. 2017;96(8):565-71.

4. Response to comment: on the results section the authors state that: “The results of subgroup analysis suggested that the effectiveness of tDCS seems to be slightly better for ischemic stroke than hemorrhagic stroke, but there was not significant difference.” However, in their conclusions they state: "tDCS is likely to be effective for patients with cognitive impairment after stroke, and the effect might be different among hemorrhagic and ischemic stroke". This conclusion doesn’t seem to be based on facts, at least from their own results.

Response: we are very sorry for the inaccurate description of the conclusion. Based on the subgroup analysis, there was actually no statistic difference between ischemic stroke and hemorrhagic stroke. And we have made correction in the conclusion section according to the facts. 

5. Response to comment: please consider revising the language specifically of the following paragraphs, in order to be able to follow the study properly:

“Except for high morbidity and mortality, the burden of stroke related-disability is another big problem in survivors, of which, the incidence of poststroke cognitive impairment (PSCI) is ranging from 22% to 47% in different studies [3-5], it has had a serious impact on both the economic and the quality of life.”

“Transcranial direct current stimulation (tDCS) was firstly developed in 1992 for

clinical purposes; nowadays, tDCS constitutes as a promising method in neurological

condition regulating.”

“Specifically, some other measurements are used for the evaluation of

attention and memory ability separately, including Computerized Neuropsychological

Test (CNT), Loewenstein occupational therapy cognitive assessment (LOTCA) and

various other forms of testing used in clinical.”

“Seven trials included two stroke types, seven studies focused on ischemic stroke and one studies on hemorrhagic stroke alone.”

Response: thanks very much for your careful checking of our manuscript. We have consulted native English speakers for manuscript revision before the submission this time and the above sentences have been revised.

---

## [Decision Letter · Decision Letter 1]

11 Feb 2020

PONE-D-19-28981R1

Effect of transcranial direct current stimulation on cognitive function in stroke patients: a systematic review and meta-analysis

PLOS ONE

Dear Dr. Liu,

Thank you for submitting your manuscript to PLOS ONE. Your paper was reevaluated by the same reviewers. Both of them indicated that the paper has improved a lot, however, there are still critical points that should be clarified. Therfore, after careful consideration, we feel that it has merit but does not fully meet PLOS ONE’s publication criteria as it currently stands. Therefore, we invite you to submit a revised version of the manuscript that addresses all points raised during the review process.

We would appreciate receiving your revised manuscript by 11th of May, 2020. To enhance the reproducibility of your results, we recommend that if applicable you deposit your laboratory protocols in protocols.io, where a protocol can be assigned its own identifier (DOI) such that it can be cited independently in the future. For instructions see: http://journals.plos.org/plosone/s/submission-guidelines#loc-laboratory-protocols

We look forward to receiving your revised manuscript.

Kind regards,

Andrea Antal, PhD

Academic Editor

PLOS ONE

Reviewers' comments:

Reviewer's Responses to Questions

**Comments to the Author**

1. If the authors have adequately addressed your comments raised in a previous round of review and you feel that this manuscript is now acceptable for publication, you may indicate that here to bypass the “Comments to the Author” section, enter your conflict of interest statement in the “Confidential to Editor” section, and submit your "Accept" recommendation.

Reviewer #1: (No Response)

Reviewer #2: All comments have been addressed

2. Is the manuscript technically sound, and do the data support the conclusions?

Reviewer #1: Partly

Reviewer #2: Yes

3. Has the statistical analysis been performed appropriately and rigorously? 

Reviewer #1: No

Reviewer #2: Yes

4. Have the authors made all data underlying the findings in their manuscript fully available?

Reviewer #1: Yes

Reviewer #2: Yes

5. Is the manuscript presented in an intelligible fashion and written in standard English?

Reviewer #1: Yes

Reviewer #2: Yes

6. Review Comments to the Author

Reviewer #1: The authors address most of the comments very well, but a still would like to bring up a few points:

-Since you changed the term "anodal tDCS" from the original version of the manuscript to "active tDCS" is now unclear to the reader, which type of tDCS has been provided in the included studies. I therefore suggest to add the missing information in table 1 (maybe in the "intervention"column), since I regarded as crucial for the reader to exactly know, to what type of tDCS you refer your analysis)

- you state that "tDCS was first developed for clinical purposes in 1992" [6]. I am not sure, if reference [6] supports this. Maybe the authors would like to refer to PMID 10990547, which is a widely accepted reference in this regard.

- Please cite review manager 5.3 in the following way, which is shown in the programme under "help_>about"

- @Risk of bias within the studies: Here you refer the corresponding biases, which I regard as discussion and would recommend to move it to the discussion section to "strength and weaknesses"

- @synthesis of results: "Thirteen studies..." I would recommend to also state the number of participants included in these 13 studies.

- @General cognitive function: you have combined MMSE points together with MoCA points in a mean difference (MD) analysis. As far as I know, this is not allowed, since these are different outcome measures for the same outcome. I therefore recommend to recalculate these analyses and all corresponding subgroup analysis as well as the assessment of publication bias with standardised mean difference (SMD) analysis. I am not sure if this will substantially affect the magnitude and direction of effect, but it would be correct.

-@Subgroup analysis: when reporting the results of subgroup analysis I would recommend to also report the results of the Chi² test for subgroup differences at the very bottom of the forest plots.

-@ summary of the evidence: you state, that there were 14 RCTs included, but shouldn't it be either 15 (for qualitative analysis) or 13 (for quantitative synthesis) as presented in figure 1? Please check.

- "Furthermore, active tDCS was beneficial for attention specifically but not for memory specifically." I would recommend to move this sentence exactly one sentence downwards in order to make clear, the following sentence about the subgroup analysis refers only to the outcome general cognitive function.

-"In addition, the other two studies excluded from the statistical pooling both reported evidence of effects in favor of active tDCS regarding measures of cognitive function." I would recommend to move this sentence to the results section, because I I would regard them as a result, which should first be presented and second be discussed.

-@Comparison with other studies: "Moreover, in 2007, the evidence-based tDCS guidelines" shouldn't it be 2017? Please check.

-@Strengths and limitations: I think a considerable strength of your work is that you have searched specifically Chinese databases and that you have got included several RCTs from China, which the authors of comparable reviews have not. Maybe you would like to state that? I think this also contributes to defect, that source 38 only has identified only one RCT regarding to your research question.

-@Implications for practice: "are most likely to benefit from transcranial" One of the weaknesses of your work is the risk of bias rating of included studies, meaning that it is likely that your effects may be biased or overestimated. Therefore I would suggest to change, i.e. to downgrade your statement a bit and mention the considerable risk of bias in the included studies

-@implications for practice:"(ischemic MD = 3.67; hemorrhagic MD = 3.12) and shorter disease duration (< 30 days MD = 3.68;

30 days -50 days MD =3.10; = 50 days MD = 1.95);" I would recommend not to state the effect sizes here in the discussion section, since I think it will confuse the reader. I think it would be better to describe it and to refer to the corresponding subgroup analysis.

-"In addition, we made an effort to divide the results by stimulation parameters, including intensity, the number of sessions and duration" Please move to the results section since these results were not presented before discussion.

-"Despite the substantial cognitive effects of active" I would downgrade this sentence in regard to the risk of bias issues, is already commented above.

-@Conclusions:What about demanding for more high-quality studies like you did in the abstract? I would support this.

Reviewer #2: I consider that the authors addressed reviewers comments appropriately. The manuscript now is written in a clear language and reflects more accurately their scientific findings.

7. PLOS authors have the option to publish the peer review history of their article (what does this mean?). If published, this will include your full peer review and any attached files.

Reviewer #1: Yes: Bernhard Elsner

Reviewer #2: Yes: Andrea G.P. Schjetnan

---

## [Author Response · Author response to Decision Letter 1]

8 May 2020

Dear Editor and Reviewers:

We are truly grateful to yours critical comments and thoughtful suggestions concerning our manuscript entitled “Effect of transcranial direct current stimulation on cognitive function in stroke patients: a systematic review and meta-analysis”( PONE-D-19-28981R1).

Based on the comments we received, careful modifications have been made to original manuscript. And all changes were marked in red text.

We hope the new manuscript will meet your magazine’s standard. Below you will find our point-by-point responses to the reviewers’ comments/ questions:

Reviewer #1: The authors address most of the comments very well, but a still would like to bring up a few points:

Suggestion 1: -Since you changed the term "anodal tDCS" from the original version of the manuscript to "active tDCS" is now unclear to the reader, which type of tDCS has been provided in the included studies. 

I therefore suggest to add the missing information in table 1 (maybe in the "intervention"column), since I regarded as crucial for the reader to exactly know, to what type of tDCS you refer your analysis)

Our modification: as suggested by the reviewers, it is important to clarify the type of tDCS included in our analysis. Therefore, we finally decided to use "anodal tDCS" to describe the intervention, just like the original version, since all the included studies used anodal stimulation and it does not lead to ambiguity. (Results section, table 1, page 9-10)

Suggestion 2: - you state that "tDCS was first developed for clinical purposes in 1992" [6]. I am not sure, if reference [6] supports this. Maybe the authors would like to refer to PMID 10990547, which is a widely accepted reference in this regard.

Our modification: thank you so much for bringing this to our attention. We reviewed the study recommended by the author and cited them in the manuscript. (Introduction section, line 43-44, page 3)

Suggestion 3: - - Please cite review manager 5.3 in the following way, which is shown in the programme under "help_>about"

Our modification: thank you very much for pointing out the problem. We changed it as “All statistical comparisons were performed in Review Manager 5.3 (http://www.ims.cochrane.org/revman/)[22].”(Materials and methods section, line 118, page 6)

Suggestion 4: - @Risk of bias within the studies: Here you refer the corresponding biases, which I regard as discussion and would recommend to move it to the discussion section to "strength and weaknesses"

Our modification: as suggested by the reviewer, we moved the “uncertainty about the reporting bias and attrition bias does exist since no priori published trial protocols for the included studies were found.” to the “strength and weaknesses” (Discussion section, line 262-263, page15)

Suggestion 5: - @synthesis of results: "Thirteen studies..." I would recommend to also state the number of participants included in these 13 studies.

Our modification: as reviewer suggested, we added the number of participants included in these 13 studies.(Results section, line 169, page11)

Suggestion 6: - @General cognitive function: you have combined MMSE points together with MoCA points in a mean difference (MD) analysis. As far as I know, this is not allowed, since these are different outcome measures for the same outcome. I therefore recommend to recalculate these analyses and all corresponding subgroup analysis as well as the assessment of publication bias with standardised mean difference (SMD) analysis. I am not sure if this will substantially affect the magnitude and direction of effect, but it would be correct.

Our modification: it’s really true as reviewer suggested that we should used the standardized mean differences (SMD) instead of the mean difference (MD) in the general cognitive function analysis, since there are two different outcome measures. We have recalculate these analyses and all corresponding subgroup analysis as well as the assessment of publication bias.

In addition, we added “Sensitivity analysis showed that Song (2019) is a major source of heterogeneity and they did not report the duration of treatment. After removing this study, the SMD for cognitive function test was 1.17 (95% CI 0.83 - 1.51, P < 0.00001, I2=78%). For this reason, we omitted this study from the subgroup analyses.” 

Then, in the following three subgroup analyses, we excluded the study of Song(2019).

(Result section, line 182-198, page 12)

Suggestion 7: -@Subgroup analysis: when reporting the results of subgroup analysis I would recommend to also report the results of the Chi² test for subgroup differences at the very bottom of the forest plots.

Our modification: as reviewer suggested, we added the results of the Chi² test and I2 value for subgroup differences. (Result section, line 190-198, page 12)

Suggestion 8: -@ summary of the evidence: you state, that there were 14 RCTs included, but shouldn't it be either 15 (for qualitative analysis) or 13 (for quantitative synthesis) as presented in figure 1? Please check.

Our modification: we are so sorry for our careless and mistakes, we checked the number of included studies and changed the “14 RCTs” to “13 studies” in the manuscript. (Discussion section, line 234, page13)

Suggestion 8: - "Furthermore, active tDCS was beneficial for attention specifically but not for memory specifically." I would recommend to move this sentence exactly one sentence downwards in order to make clear, the following sentence about the subgroup analysis refers only to the outcome general cognitive function.

Our modification: as suggested by the reviewer, we moved this sentence one sentence downwards.(Discussion section, line 235-238, page14)

Suggestion 9: -"In addition, the other two studies excluded from the statistical pooling both reported evidence of effects in favor of active tDCS regarding measures of cognitive function." I would recommend to move this sentence to the results section, because I would regard them as a result, which should first be presented and second be discussed. 

Our modification: as suggested by the reviewer, we moved this sentence to the result section. (Results section, line 219-220, page 13)

Suggestion 10: -@Comparison with other studies: "Moreover, in 2007, the evidence-based tDCS guidelines" shouldn't it be 2017? Please check.

Our modification: we are very sorry for our careless and mistakes, we’ve changed it to “2017” in manuscript.(Discussion section, line 250, page 14)

Suggestion 10: -@Strengths and limitations: I think a considerable strength of your work is that you have searched specifically Chinese databases and that you have got included several RCTs from China, which the authors of comparable reviews have not. Maybe you would like to state that? I think this also contributes to defect, that source 38 only has identified only one RCT regarding to your research question. 

Our modification: thank you very much for bringing this to our attention, we added “we conducted an extensive search of the Chinese database, so we obtained several recently published RCTs” and “73% of the included studies came from China and there was a lack of English studies” in the Strengths and limitations part. (Discussion section, line 255-259, page 14-15)

Suggestion 11: -@Implications for practice: "are most likely to benefit from transcranial" One of the weaknesses of your work is the risk of bias rating of included studies, meaning that it is likely that your effects may be biased or overestimated. Therefore I would suggest to change, i.e. to downgrade your statement a bit and mention the considerable risk of bias in the included studies

Our modification: it is really true as reviewer suggested that we should downgrade the statement, Considering the risk of bias in the included study. 

We have modified this part as:

This meta-analysis suggest that stroke patients with cognitive function deficits appear to benefit from transcranial direct current stimulation. However, on the basis of this review of the published literature and considering the risk of bias of included studies, tDCS cannot yet be recommended as a standard therapy for the stokes. More research is needed to determine the potential benefits of tDCS in the future. (Discussion section, line 269-273, page 15)

Suggestion 12: -@implications for practice:"(ischemic MD = 3.67; hemorrhagic MD = 3.12) and shorter disease duration (< 30 days MD = 3.68;

30 days -50 days MD =3.10; = 50 days MD = 1.95);" I would recommend not to state the effect sizes here in the discussion section, since I think it will confuse the reader. I think it would be better to describe it and to refer to the corresponding subgroup analysis.

Our modification: we rewrote this paragraph due to slight change of the results, and as suggested by the reviewer, we removed the effect sizes in the discussion section and used corresponding subgroups to describe it. (Discussion section, line 280-286, page 15-16)

Suggestion 13: -"In addition, we made an effort to divide the results by stimulation parameters, including intensity, the number of sessions and duration" Please move to the results section since these results were not presented before discussion.

Our modification: considering the reviewer’s suggestion, we removed the sentences “In addition, we made an effort to divide the results by stimulation parameters, including intensity, the number of sessions and duration. However, no differences were found in these comparisons, possibly due to the heterogeneity of the remaining parameters.” from the discussion section. 

However, this subgroup analysis results were not added to the results section, because we find that it is unreasonable to carry out subgroup analysis on only one stimulus parameter without controlling other stimulus variables including intensity, the number of sessions and duration.

Suggestion 14: -"Despite the substantial cognitive effects of active" I would downgrade this sentence in regard to the risk of bias issues, is already commented above.

Our modification: thank you for reminding us to correct this mistake again, we changed this sentence to “Although current studies have shown that tDCS may be effective on cognitive function recovery after stroke”. (Discussion section, line 291-292, page 17)

Suggestion 15: -@Conclusions:What about demanding for more high-quality studies like you did in the abstract? I would support this.

Our modification: thank you very much for these valuable suggestions, We revised the last two sentences of the conclusion into: “However, the evidence on the impact of patient characteristics and stimulation parameters is still lacking. More high-quality research is needed to determine the efficacy of tDCS in the treatment of cognitive deficits after stroke and to establish the optimal treatment program.” (Conclusions section, line 303-306, page17)

Reviewer #2: I consider that the authors addressed reviewers comments appropriately. The manuscript now is written in a clear language and reflects more accurately their scientific findings.

Special thanks to you for your good comments.

---

## [Decision Letter · Decision Letter 2]

15 May 2020

Effect of transcranial direct current stimulation on cognitive function in stroke patients: a systematic review and meta-analysis

PONE-D-19-28981R2

Dear Dr. Liu,

We are pleased to inform you that your manuscript has been judged scientifically suitable for publication and will be formally accepted for publication once it complies with all outstanding technical requirements.

With kind regards,

Andrea Antal, PhD

Academic Editor

PLOS ONE

Additional Editor Comments (optional):

Reviewers' comments:

Reviewer's Responses to Questions

**Comments to the Author**

1. If the authors have adequately addressed your comments raised in a previous round of review and you feel that this manuscript is now acceptable for publication, you may indicate that here to bypass the “Comments to the Author” section, enter your conflict of interest statement in the “Confidential to Editor” section, and submit your "Accept" recommendation.

Reviewer #1: All comments have been addressed

Reviewer #2: All comments have been addressed

2. Is the manuscript technically sound, and do the data support the conclusions?

Reviewer #1: Yes

Reviewer #2: Yes

3. Has the statistical analysis been performed appropriately and rigorously? 

Reviewer #1: Yes

Reviewer #2: Yes

4. Have the authors made all data underlying the findings in their manuscript fully available?

Reviewer #1: Yes

Reviewer #2: Yes

5. Is the manuscript presented in an intelligible fashion and written in standard English?

Reviewer #1: Yes

Reviewer #2: Yes

6. Review Comments to the Author

Reviewer #1: (No Response)

Reviewer #2: I consider that the authors addressed the valuable suggestions of Reviewer #1 appropriately. I think this has improved the article significantly and, in my view is ready to be accepted.

7. PLOS authors have the option to publish the peer review history of their article (what does this mean?). If published, this will include your full peer review and any attached files.

Reviewer #1: Yes: Bernhard Elsner

Reviewer #2: Yes: Andrea Gomez Palacio Schjetnan

---

## [Editor Report · Acceptance letter]

21 May 2020

PONE-D-19-28981R2 

Effect of transcranial direct-current stimulation on cognitive function in stroke patients: a systematic review and meta-analysis 

Dear Dr. Liu:

I am pleased to inform you that your manuscript has been deemed suitable for publication in PLOS ONE. Congratulations! Your manuscript is now with our production department. 

With kind regards,

on behalf of

Prof. Dr. Andrea Antal 

Academic Editor

PLOS ONE